# Morolic Acid 3-*O*-Caffeate Inhibits Adipogenesis by Regulating Epigenetic Gene Expression

**DOI:** 10.3390/molecules25245910

**Published:** 2020-12-13

**Authors:** Sook In Chae, Sang Ah Yi, Ki Hong Nam, Kyoung Jin Park, Jihye Yun, Ki Hyun Kim, Jaecheol Lee, Jeung-Whan Han

**Affiliations:** 1School of Pharmacy, Sungkyunkwan University, Suwon 16419, Korea; tmrdlsdl7@naver.com (S.I.C.); angelna1023@hanmail.net (S.A.Y.); nam6422@hanmail.net (K.H.N.); taciturnsoul@naver.com (K.J.P.); wlgpp405@gmail.com (J.Y.); khkim83@skku.edu (K.H.K.); jaecheol@skku.edu (J.L.); 2Biomedical Institute for Convergence at SKKU (BICS), Sungkyunkwan University, Suwon 16419, Korea; 3Imnewrun Biosciences Inc., Suwon 16419, Korea

**Keywords:** obesity, adipogenesis, morolic acid 3-*O*-caffeate, *Betula schmidtii*, *Wnt6*, histone H3 lysine 4 trimethylation

## Abstract

Obesity causes a wide range of metabolic diseases including diabetes, cardiovascular disease, and kidney disease. Thus, plenty of studies have attempted to discover naturally derived compounds displaying anti-obesity effects. In this study, we evaluated the inhibitory effects of morolic acid 3-*O*-caffeate (MAOC), extracted from *Betula schmidtii*, on adipogenesis. Treatment of 3T3-L1 cells with MAOC during adipogenesis significantly reduced lipid accumulation and decreased the expression of adiponectin, a marker of mature adipocytes. Moreover, the treatment with MAOC only during the early phase (day 0–2) sufficiently inhibited adipogenesis, comparable with the inhibitory effects observed following MAOC treatment during the whole processes of adipogenesis. In the early phase of adipogenesis, the expression level of *Wnt6,* which inhibits adipogenesis, increased by MAOC treatment in 3T3-L1 cells. To identify the gene regulatory mechanism, we assessed alterations in histone modifications upon MAOC treatment. Both global and local levels on the *Wnt6* promoter region of histone H3 lysine 4 trimethylation, an active transcriptional histone marker, increased markedly by MAOC treatment in 3T3-L1 cells. Our findings identified an epigenetic event associated with inhibition of adipocyte generation by MAOC, suggesting its potential as an efficient therapeutic compound to cure obesity and metabolic diseases.

## 1. Introduction

Obesity is caused by the accumulation of excessive amounts of fat in adipose tissues in the body due to an imbalance between food intake and energy use over a prolonged period [1]. This prolonged overnutrition induces de novo adipocyte differentiation from preadipocytes, which contributes adipose tissue expansion [2]. Hence, there have been many efforts to identify bioactive compounds that block adipogenesis for the treatment of obesity.

*Betula schmidtii,* which belongs to the Betulaceae family, contains diverse bioactive compounds including phenolic derivatives [3] and triterpenoids [4]. Among the bioactive substances isolated from *Betula schmidtii*, morolic acid 3-*O*-caffeate (MAOC) is one of oleanane-type triterpenoid derivatives (Figure 1A). It has been reported that diverse oleanane triterpenoids, including oleanolic acid, betulinic acid, and ursolic acid, exhibit anti-obesity effects. Oleanolic acid is known to suppress adipogenesis of 3T3-L1 preadipocytes [5]. Additionally, ursolic acid also blocks adipogenesis of 3T3-L1 cells via the liver kinase B1 (LKB1)/AMP-activated protein kinase (AMPK) signaling pathway [6] and improves insulin resistance and hyperinsulinemia in obese rats [7]. Another oleanane triterpenoid, betulinic acid, has various pharmacological effects, including apoptotic [8] and antiviral [9] activities. Betulinic acid also shows anti-obesity effects, reducing body weight in a diet-induced obese mouse model [10], as well as lowers lipid accumulation in 3T3-L1-derived adipocytes [11]. Despite these metabolic advantages of oleanane triterpenoids, the effects of morolic acid or its derivatives on obesity have been hardly investigated.

In our previous study, we reported that morolic acid 3-*O*-caffeate (MAOC), a triterpenoid derived from *B. schmidtii*, exhibits potent cytotoxic activities against cancer cells [4]. In this current study, we evaluated the effects of MAOC isolated from *B. schmidtii* on adipocyte differentiation from preadipocytes. We stained lipid accumulated in mature adipocytes by Oil Red O staining and measured the expression of adipocyte-specific marker genes in adipocytes treated with MAOC during the differentiation period. Furthermore, we revealed the epigenetic alteration that is involved in the anti-adipogenic action of MAOC. These results proved that MAOC is a promising candidate for anti-obesity drug development.

## 2. Results and Discussion

### 2.1. Viability of 3T3-L1 Cells Treated with Morolic Acid 3-O-Caffeate

Based on the fact that ursolic acid, oleanolic acid, and betulinic acid, which have structures similar to morolic acid, exhibit anti-adipogenic activity [5,6,11], we investigated the effects of MAOC on adipogenesis. To determine the treatment concentration of MAOC to 3T3-L1 cells, we first assessed the cytotoxicity of MAOC in 3T3-L1 preadipocytes. 3T3-L1 cells were treated with MAOC at various concentrations (0, 2.5, 5, 10, and 20 μM) for 24 h. Cytotoxic effects were not observed at 2.5 and 5 μM, but higher doses (10 and 20 μM) of MAOC significantly decreased the viability of 3T3-L1 cells (Figure 1B). Thus, 5 μM of MAOC was used for the further experiments to assess the anti-adipogenic effects of it in 3T3-L1 cells.

### 2.2. Inhibitory Effects of Morolic Acid 3-O-Caffeate on Adipogenesis

Next, we evaluated the effects of MAOC on the differentiation into mature adipocytes from 3T3-L1 preadipocytes by treating 3T3-L1 cells with MAOC during the entire period of adipogenesis (Figure 2A). After maturation, accumulated lipid droplets within mature adipocytes were visualized by Oil Red O staining (Figure 2B). MAOC treatment markedly decreased the number of adipocytes, especially at 5 and 10 μM (Figure 2B). Next, we assessed the mRNA level of adipocyte marker genes (*Adipsin* and *Adipoq*) using reverse transcription–quantitative PCR (RT-qPCR). The transcription levels of both genes were significantly reduced when the cells were treated with 5 or 10 μM of MAOC during adipogenesis (Figure 2C). Furthermore, we observed a decrease in the protein level of adiponectin, a specific marker for functional adipocytes, using Western blot assay (Figure 2D). These data demonstrate that MAOC exhibits inhibitory effects on adipocyte differentiation.

Adipogenesis from preadipocytes is a multistep process that accompanies the sequential activation of several signaling pathways and essential transcription factors [12]. During the early stages (days 0–2), a proliferative burst termed mitotic clonal expansion (MCE) [13] and a drastic reduction in Wnt signaling [14] can be observed. From middle to late stages of adipogenesis (days 2–8), the key transcription factors, CCAAT/enhancer-binding protein (C/EBP) gene family and peroxisome proliferator activated receptor-γ (PPAR-γ) induce terminal differentiation in a cooperative manner [15]. To investigate which stage of adipogenesis is affected by MAOC, we treated 3T3-L1 cells with MAOC in the early (day 0–2) and late (day 2–8) stages of adipogenesis (Figure 3A). Oil Red O staining data revealed that MAOC treatment during the first 2 days decreased adipocyte generation as well as accumulation of lipid droplets, whereas MAOC treatment during the late stage failed to induce considerable changes (Figure 3B). Consistently, treatment with MAOC during the early stages decreased the mRNA expression of adipocyte marker genes (*Adipsin* and *Adipoq*) to a level comparable with that observed following MAOC treatment during the entire process of adipogenesis (Figure 3C). Furthermore, expression of adiponectin was also reduced upon exposure to MAOC during the early stage as well as during the entire period of adipogenesis (Figure 3D). These results indicate that the early stages of adipogenesis were impaired by MAOC.

### 2.3. Epigenetic Activation of Wnt6 Expression by Morolic Acid 3-O-Caffeate

It has been proposed that adipogenesis is epigenetically regulated by histone methylation [16]; trimethylation of histone H3 lysine 4 (H3K4me3) activates the expression of adipogenic transcription factors [17], while the trimethylation of histone H3 lysine 27 (H3K27me3) suppresses the expression of Wnt ligands that inhibit adipogenesis [18,19]. Thus, we examined the global level of the two histone modifications, H3K4me3 and H3K27me3, after treating 3T3-L1 cells with MAOC (Figure 4A). Upon MAOC treatment, the global level of H3K4me3 was elevated, while that of H3K27me3 was not affected by MAOC (Figure 4A). Earlier studies have demonstrated that the expression of *Wnt6*, *Wnt10a*, and *Wnt10b*, known inhibitors of adipogenesis, drastically decreased during the early stage of adipogenesis [14]. Among the three genes, *Wnt6* is known to be upregulated by H3K4me3 when histone demethylase KDM5A is depleted during adipogenesis [20]. Therefore, we aimed to determine whether the anti-adipogenic effects of MAOC are mediated via the upregulation of *Wnt6* expression. The RT-qPCR analysis showed that MAOC treatment during adipogenesis markedly increased the mRNA expression of *Wnt6* gene (Figure 4B). Moreover, Chromatin immunoprecipitation (ChIP)-qPCR analysis showed that enrichment of H3K4me3 on *Wnt6* promoter region was significantly elevated upon MAOC treatment, while H3 occupancy remained unaltered (Figure 4C). These data demonstrate that MAOC treatment inhibits adipogenesis by promoting *Wnt6* transcription by increasing the H3K4me3 level on its promoter.

During the development of obesity, surplus energy promotes the generation of adipocytes from precursor cells, which can provide an additional reservoir to store excess body fat [1,2]. Therefore, pharmacological approaches to inhibit adipogenesis have been considered useful strategies to ameliorate diverse symptoms of metabolic diseases. Our current study identified that MAOC, obtained from *B. schmidtii*, can effectively disrupt adipogenesis from preadipocytes. Moreover, we here elucidated the molecular changes related to the anti-adipogenic effect of this compound (Figure 5). Adipocyte differentiation is finely controlled through a cooperative network of active and repressive histone markers [16]. We demonstrated that treatment with MAOC enhanced the global level of H3K4me3, an active histone marker. Furthermore, we observed the increase in H3K4me3 levels on the promoter region of *Wnt6*, which subsequently increased its transcription. As a drastic reduction of *Wnt6* expression during the early stage is required for adipogenesis from 3T3-L1 cells [14], MAOC treatment only during the early stage could successfully disrupt adipocyte differentiation from preadipocytes.

The possible implication of MAOC in the clinical usages remains unclear, because MAOC has not been administered to animal model or patients. Given that MAOC exhibits anti-inflammatory and radical scavenging activity in vitro [21,22], MAOC can display protective effects in preclinical or clinical studies. Thus, it appears that MAOC can be a feasible option to treat obesity and metabolic complications.

## 3. Materials and Methodsα

### 3.1. Isolation of Morolic Acid 3-O-Caffeate (MAOC)

MAOC was obtained from *Betula schmidtii* twigs as previously described [4]. Briefly, MAOC was extracted from *B. schmidtii* twigs using 80% aqueous MeOH (Samchun chemicals, Pyeongtaek, Korea) for 1day under reflux. The resultant MeOH extract (410 g) was suspended in water (2.4 L) and then successively partitioned with *n*-hexane (Samchun chemicals, Pyeongtaek, Korea), CHCl_3_ (Samchun chemicals, Pyeongtaek, Korea), EtOAc (Samchun chemicals, Pyeongtaek, Korea), and *n*-BuOH (Samchun chemiclas, Pyeongtaek, Korea), yielding 15, 18, 19, and 116 g, respectively. The CHCl_3_-soluble fraction (10 g) was subjected to a silica gel column eluted with CHCl_3_-MeOH [40:1, 30:1, 15:1, 9:1, 4:1 and 1:1] to afford six fractions (Fr. A–F). Fr. D (2 g) was fractionated into nine subfractions (Fr. D1–D9) by an RP-C_18_ silica gel open column with 60% aqueous MeOH. MAOC (11 mg) was isolated from Fr. D9 (87 mg) by semi-preparative reversed-phase HPLC using 90% aqueous MeOH. The structure of MAOC was identified by comparing the nuclear magnetic resonance (NMR) spectroscopic data with previously reported data [23], and electrospray ionization mass spectrometry (ESIMS) data.

### 3.2. Cell Culture and Differentiation

3T3-L1 preadipocytes were maintained in Dulbecco’s Modified Eagle Medium (DMEM) (Welgene, Seoul, Korea) supplemented with 10% bovine calf serum (BCS) (Welgene, Seoul, Korea)and 1% penicillin/streptomycin (P/S) (Welgene, Seoul, Korea). To induce adipocyte differentiation, 3T3-L1 cells were incubated in adipogenic media consisting of DMEM, 10% fetal bovine serum (FBS) (Welgene, Seoul, Korea), 1% P/S, 10 μg/mL insulin (Sigma-Aldrich, St.Louis, MO, USA), 0.5 mM 3-isobuyl-1-methylxanthine (IBMX) (Sigma-Aldrich, St.Louis, MO, USA) and 1 μM dexamethasone (Sigma-Aldrich, St.Louis, MO, USA). Then, media were changed every 2 days with DMEM containing 10% FBS, 1% P/S, and 10 μg/mL insulin. In our experiments, to assess the effects of MAOC on adipogenesis, the cells were treated with MAOC during adipogenesis. Eight days after initiating adipogenesis, the cells were prepared for further experiments, including RT-qPCR and Western blotting.

### 3.3. Cell Counting

3T3-L1 preadipocytes seeded on 12-well plates were treated with various concentrations of MAOC. After 24 h, the cells were detached from the plate with ethylenediaminetetraacetic acid (EDTA) and diluted with phosphate-buffered saline (PBS) (Welgene, Seoul, Korea). The number of cells was counted using LUNA-II™ Automated Cell Counter (Logos Biosystems, Anyang, Korea).

### 3.4. Oil Red O Staining

To visualize lipid droplets accumulated in adipocytes, Oil Red O staining was performed as previously described [24]. Briefly, fully differentiated adipocytes were fixed with 10% formaldehyde (Sigma-Aldrich, St.Louis, MO, USA) for 10 min and washed with 60% isopropanol (Sigma-Aldrich, St.Louis, MO, USA). Then, the Oil Red O working solution was added to each well for 1 h, followed by washing with distilled water. The stained lipid droplets were captured by Cytation^TM^ 5 Cell Imaging Multi-Mode Reader (Bio Tek, Winooski, VT, USA).

### 3.5. Reverse Transcription (RT) and Quantitative Real-Time PCR (qPCR)

To measure the transcription level of genes, total RNA was extracted with Easy-Blue reagent (Intron Biotechnology, Seongnam, Korea). The reaction of reverse transcription was performed using a Maxim RT-PreMix Kit (Intron Biotechnology, Seongnam, Korea). For quantitative real-time PCR (qPCR), the cDNA was mixed with KAPA SYBR^®^ FAST qPCR Master Mix (Kapa Biosystems, Wilmington, MA, USA), and the PCR reaction was detected with a CFX96 Touch™ real-time PCR detector (Bio-Rad, Hercules, CA, USA). Relative mRNA levels of each reaction were normalized to the mRNA levels of *β-actin*. The PCR primer sequences used are as follows: *β-actin* forward, 5′-ACGGCCAGGTCATCACTATTG-3′, *β-actin* reverse, 5′-TGGATGCCACAGGATTCCA-3′, *Adipsin* forward, 5′-CATGCTCGGCCCTACATG-3′, *Adipsin* reverse, 5′-CACAGAGTCGTCATCCGTCAC-3′, *Adipoq* forward, 5′-TGTTCCTCTTAATCCTGCCCA-3′, *Adipoq* reverse, 5′-CCAACCTGCACAAGTTCCCTT-3′, *Wnt6* forward 5′-GCGGAGACGATGTGGACTTC-3′, *Wnt6* reverse, 5′-ATGCACGGATATCTCCACGG-3′.

### 3.6. Western Blot

The Western blot assay was performed as previously described [25]. Proteins were extracted from cells using Pro-Prep (Intron Biotechnology) followed by sonication and centrifugation (12,000 rpm, 15 min, 4 °C). Extracted proteins were separated by sodium dodecyl sulfate (SDS)-polyacrylamide gel (12%) electrophoresis. Then, the proteins were transferred to polyvinylidene difluoride (PVDF) membranes using wet transfer (Bio-Rad, USA). The membranes were incubated with primary antibodies overnight at 4 °C, and then incubated with horseradish peroxidase (HRP)-conjugated secondary antibodies (Abcam, Cambridge, UK) for 1 h at room temperature. The signals from the reaction with a chemiluminescence reagent (Abclon, Guro, Korea) were detected on the AGFA X-ray film Cp-Bu NEW (Agfa-Gevaert, Mortsel, Belgium). Anti-α-tubulin (Santa Cruz Biotechnology, Dallas, TX, USA; SC-32293), anti-actin (Merck Millipore, Burlington, MA, USA; mab1501), anti-adiponectin (Cell Signaling Technology, Danvers, MA, USA; 2789), anti-histone H3 (Santa Cruz Biotechnology, SC-10809), anti-histone H3 Lys4 trimethylation (Merck Millipore; 07-473), and anti-histone H3 Lys27 trimethylation (Merck Millipore; 07-449) were used for the Western blot in this study.

### 3.7. Chromatin Immunoprecipitation (ChIP)-qPCR

To evaluate the enrichment of histone modification on the *Wnt6* gene promoter, ChIP-qPCR analysis was performed as previously demonstrated [26]. In brief, 3T3-L1 cells were fixed with 1% formaldehyde for 15 min, followed by sonication. The sheared chromatin was obtained by centrifugation (13,000 rpm, 20 min, 4 °C). A small portion (5%) of the chromatin solution was reserved as input DNA, and the remaining chromatin solution was incubated with the primary antibodies overnight at 4 °C. Then, the immunoprecipitated chromatin fragments were reverse-crossed from the proteins and eluted for qPCR with primers for the promoter region of *Wnt6* gene (forward, 5′-CTTCCTTCCCCCAAAGAAATG-3′; reverse, 5′-GTCCAACAGCTCTTCCCTACCTATC-3′). Anti-histone H3 (Santa Cruz Biotechnology, SC-10809) and anti-histone H3 Lys4 trimethylation (Merck Millipore; 07-473) were used for ChIP reaction.

### 3.8. Statistical Analysis

The significance of data was determined with a two-tailed Student’s t-test using Microsoft Excel. Significance was evaluated based on the *p*-value. 

## 4. Conclusions

In this study, we identified the anti-adipogenic effects of MAOC that was isolated from *B. schmidtii*. MAOC inhibited adipocyte differentiation from preadipocytes, preventing lipid accumulation and the expression of adipocyte marker genes. Moreover, our findings elucidated that MAOC increases the levels of H3K4me3, an active histone marker, thus enhancing the expression of Wnt6, an adipogenesis-inhibiting ligand. Collectively, our findings provide experimental evidence for the use of an active triterpenoid derivative to block excessive adipogenesis in obesity.

## Figures and Tables

**Figure 1 molecules-25-05910-f001:**
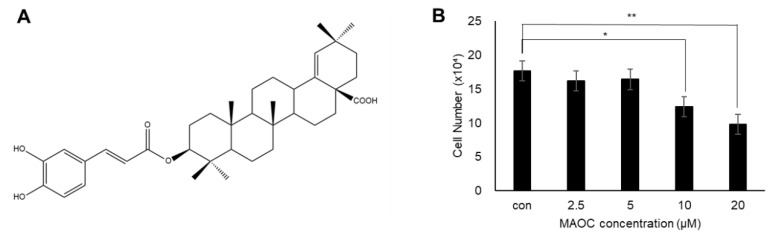
Cytotoxic effects of morolic acid 3-*O*-caffeate (MAOC) on 3T3-L1 preadipocytes. (**A**) Chemical structure of MAOC. (**B**) The number of 3T3-L1 cells treated with MAOC (0, 2.5, 5, 10, and 20 μM) for 24 h were measured. Data present the means ± SD for *n* = 3. * *p* < 0.05 and ** *p* < 0.01.

**Figure 2 molecules-25-05910-f002:**
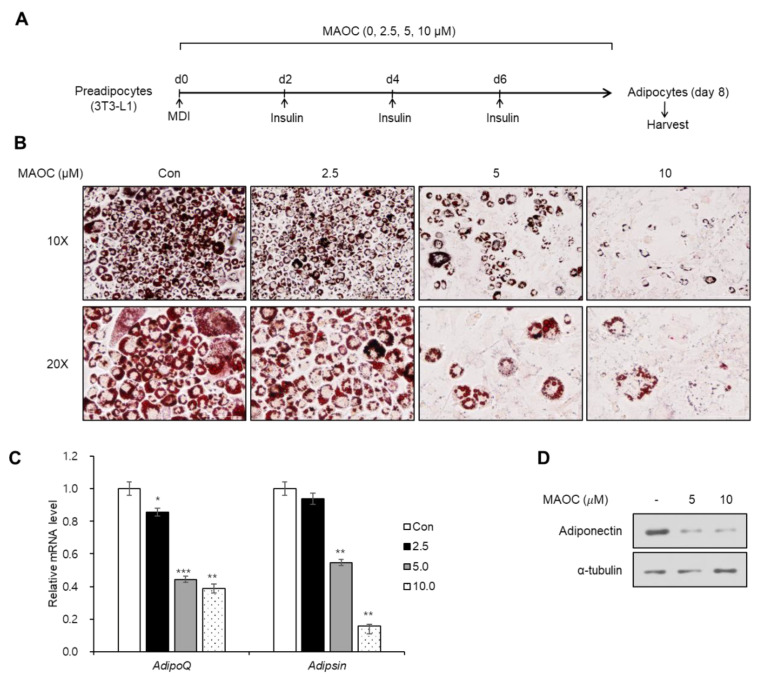
The inhibitory effects of morolic acid 3-*O*-caffeate (MAOC) on adipogenesis. (**A**) Schematic representation of 3T3-L1 differentiation into adipocytes. Cells were treated with MAOC during the entire period of differentiation. (**B**) Oil Red O staining of 3T3-L1 adipocytes incubated with MAOC during adipogenesis. (**C**) The mRNA levels of *Adipoq* and *Adipsin* genes in 3T3-L1 adipocytes incubated with MAOC during adipogenesis. Data present the means ± standard error of the mean (SEM) for *n* = 3. * *p* < 0.05, ** *p* < 0.01, and *** *p* < 0.001. (**D**) Western blot analysis of 3T3-L1 adipocytes incubated with MAOC during adipogenesis.

**Figure 3 molecules-25-05910-f003:**
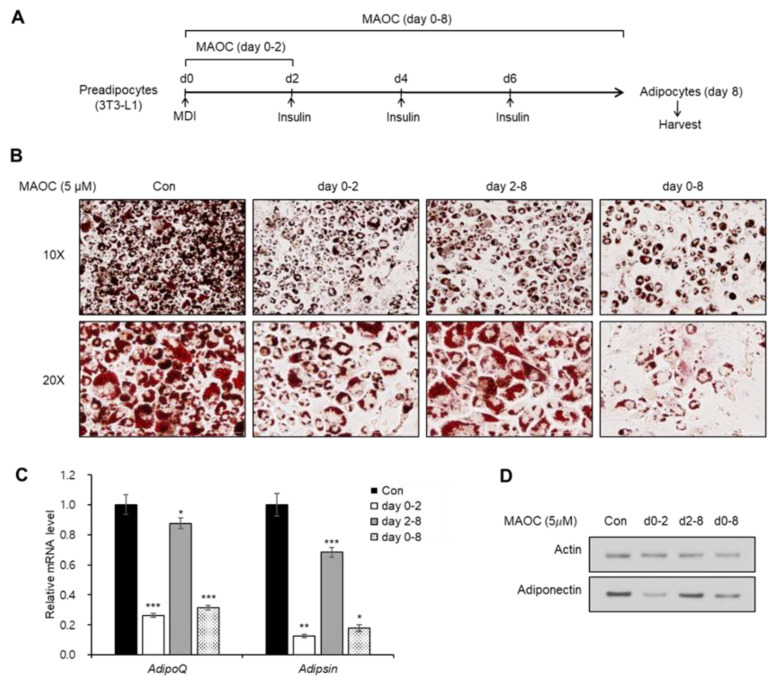
Inhibitory effects of morolic acid 3-*O*-caffeate (MAOC) on the early stage of adipogenesis. (**A**) Schematic representation of 3T3-L1 differentiation into adipocytes. Cells were treated with MAOC (5 μM) during the early days (day 0–2), late days (day 2–8), or entire period of differentiation. (**B**) Oil Red O staining of 3T3-L1 adipocytes incubated with MAOC (5 μM) during adipogenesis. (**C**) The mRNA levels of *Adipoq* and *Adipsin* genes in 3T3-L1 adipocytes incubated with MAOC (5 μM) during adipogenesis. Data present the means ± SEM for *n* = 3. * *p* < 0.05, ** *p* < 0.01, and *** *p* < 0.001. (**D**) Western blot analysis of 3T3-L1 adipocytes incubated with MAOC (5 μM) during adipogenesis.

**Figure 4 molecules-25-05910-f004:**
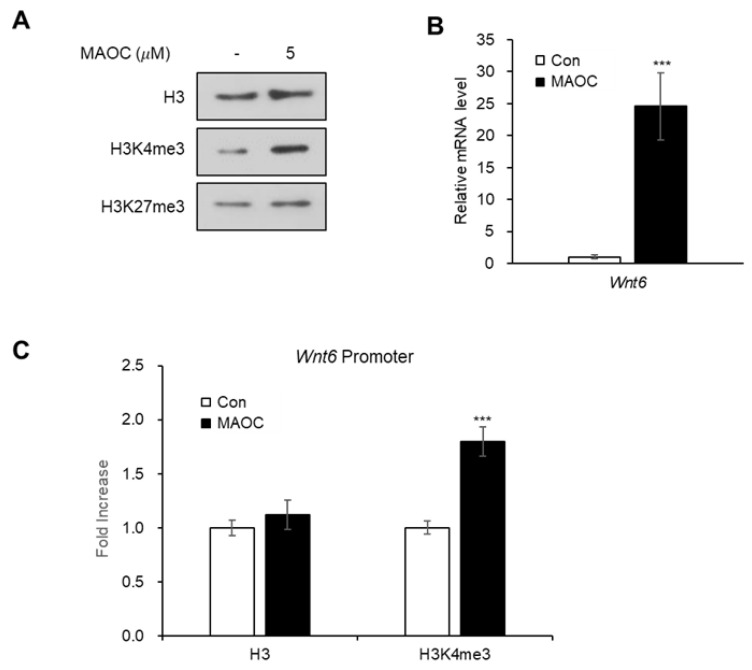
Treatment with morolic acid 3-*O*-caffeate (MAOC) increases *Wnt6* expression by enhancing H3K4me3 enrichment on promoter. (**A**) Western blot analysis of 3T3-L1 adipocytes incubated with MAOC (5 μM) during adipogenesis. (**B**) The mRNA levels of *Wnt6* gene in 3T3-L1 adipocytes incubated with MAOC (5 μM) during adipogenesis. Data present the means ± SEM for *n* = 3. *** *p* < 0.001. (**C**) 3T3-L1 cells were treated with MAOC (5 μM) for 24 h. ChIP assay was performed with H3 and H3K4me3 antibodies followed by qPCR with primers for promoter region of *Wnt6* gene. Data present the means ± SEM for *n* = 3. *** *p* < 0.001.

**Figure 5 molecules-25-05910-f005:**
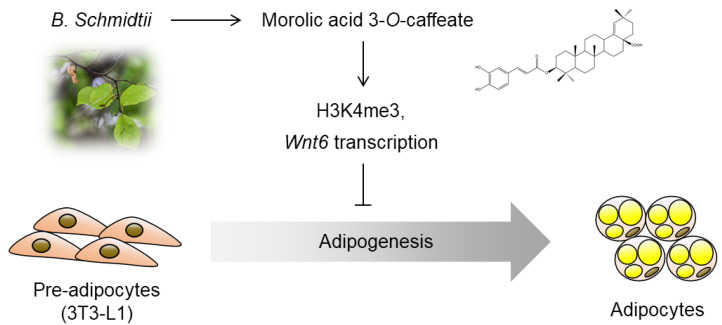
Molecular changes related to the anti-adipogenic action of morolic acid 3-*O*-caffeate (MAOC). MAOC inhibits adipocyte differentiation from 3T3-L1 preadipocytes by enhancing active histone mark (H3K4me3) and transcription of *Wnt6* expression, which is one of adipogenesis-inhibiting ligands.

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
