# Peer review of "Morolic Acid 3-O-Caffeate Inhibits Adipogenesis by Regulating Epigenetic Gene Expression"

_molecules, 2020, doi:10.3390/molecules25245910_

Round 1

Reviewer 1 Report

The manuscript described the effect of Morolic Acid 3-O-Caffeate against adipogenesis in vitro, of which the underlying mechanism could involve the epigenetic regulation. The study is well designed and written in great flow. One minor suggestion would be to include some detailed discussions on potential effect in preclinical/clinical studies. The author mentioned that MAOC has shown anti-cancer properties, are the protective mechanism sharing similar pathway as in this paper. I think including these information would help to further justify the use of MAOC for future study.  

Author Response

The manuscript described the effect of Morolic Acid 3-O-Caffeate against adipogenesis in vitro, of which the underlying mechanism could involve the epigenetic regulation. The study is well designed and written in great flow. One minor suggestion would be to include some detailed discussions on potential effect in preclinical/clinical studies. The author mentioned that MAOC has shown anti-cancer properties, are the protective mechanism sharing similar pathway as in this paper. I think including these information would help to further justify the use of MAOC for future study. 

--> We thank the reviewer for recognizing the potential importance of our findings and recommending a good way to improve our manuscript. We added more detailed discussions on potential implications of MAOC in preclinical or clinical studies (lines 155-158 in revised manuscript). Accordingly, we added two more references (ref. 21 and 22 in revised manuscript) for the demonstration of physically protective effects of MAOC.

Reviewer 2 Report

This manuscript described a series of experiments which support inhibitory role of Morolic acid 3-O-caffeate obtained (MAOC) fraction obtained from B. Schmidii on adipocyte differentiation of 3T3-L1 cells. Overall, there was no major problem recognized in this manuscript. There are only a few text changes suggested.

  1. Some descriptions indicating histone methylation as direct mechanism of MAOC action were misleading. Data presented in this study only supports correlation/association. Therefore, some sentences should be modified to avoid overstatement. For examples:

Line 27: epigenetic mechanism by which > epigenetic events associated with inhibition of adipocyte generation by MAOC.

Line 56, 147, 157: we revealed epigenetic mechanism underlying - these seem to be too strong expression.

Fig 5 should use "H3K4me3, Wnt6 transcription" as it is unclear if K4me3 regulation is a cause of Wnt6 upregulation.

  1. ESIMS = Electrospray Ionization MS?
  2. Fig 1B labeling looks like uM, change to "micro" symbol?

Author Response

This manuscript described a series of experiments which support inhibitory role of Morolic acid 3-O-caffeate obtained (MAOC) fraction obtained from B. Schmidii on adipocyte differentiation of 3T3-L1 cells. Overall, there was no major problem recognized in this manuscript. There are only a few text changes suggested.

We thank the reviewer for recognizing the potential importance of our findings. We feel our responses to his/her criticisms have improved the manuscript with several corrections. Our responses to the reviewer’s queries are described point-by-point below.

Some descriptions indicating histone methylation as direct mechanism of MAOC action were misleading. Data presented in this study only supports correlation/association. Therefore, some sentences should be modified to avoid overstatement. For examples:

Line 27: epigenetic mechanism by which > epigenetic events associated with inhibition of adipocyte generation by MAOC.

We corrected the sentence, as the referee suggested (line 27 in revised manuscript).

Line 56, 147, 157: we revealed epigenetic mechanism underlying - these seem to be too strong expression.

We agree that the sentences in line 56, 147, and 157 need to be toned down. We changed ‘epigenetic mechanism underlying’ in line 56 to ‘epigenetic alteration which is involved in’. We also corrected ‘molecular mechanism underlying’ in line 147 and 157 (line 161 in revised manuscript) to ‘molecular changes related with’.

Fig 5 should use "H3K4me3, Wnt6 transcription" as it is unclear if K4me3 regulation is a cause of Wnt6 upregulation.

As the referee mentioned, it is unclear whether the increase in H3K4me3 is the major cause of Wnt6 upregulation upon MAOC treatment. Thus, we removed the arrow between “H3K4me3” and “Wnt6 transcription” in figure 5, as the referee recommended. Furthermore, we toned down the figure legend of Figure 5 (line 163 in revised manuscript).

ESIMS = Electrospray Ionization MS?

Yes, ESIMS is an abbreviation of electrospray ionization mass spectrometry. To clarify the meaning, we added the full name of ESIMS (line 176 in revised manuscript).

Fig 1B labeling looks like uM, change to "micro" symbol?

We thank the reviewer for pointing out the error. We corrected it to ‘micro’ symbol.